# Comparative Analysis of the Effects of Incorporating Post-Industrial Recycled LLDPE and Post-Consumer PE in Films: Macrostructural and Microstructural Perspectives in the Packaging Industry

**DOI:** 10.3390/polym16070916

**Published:** 2024-03-27

**Authors:** Ricardo Ballestar de las Heras, Xavier Colom, Javier Cañavate

**Affiliations:** 1Research Department of Sphere Group Spain, P.I El Pradillo 3 C/Sphere, Parcela 9, 50690 Pedrola, Zaragoza, Spain; ricardo.luis.ballestar@upc.edu; 2Chemical Engineering Department, Universitat Politècnica de Catalunya BarcelonaTECH, ESEIAAT, Colom 1, 08222 Terrassa, Barcelona, Spain; francisco.javier.canavate@upc.edu

**Keywords:** mechanical recycling, polymer degradation, circular economy, recycled, thermal analysis, FTIR, polyethylene, plastic bags

## Abstract

In accordance with the Circular Economy Package of the European directive, the Spanish government compels manufacturers of plastic bags to include into their products a minimum of 70% of polyethylene (PE) waste. Following this mandate can be challenging and requires a deep knowledge of the alterations produced by the recycling in the main components of a plastic bag film: lineal low-density polyethylene (LLDPE), the LLDPE recycled post-industry, generated as waste from an industrial process (rLLDPE) and the PE recycled from post-consumer use (rPE), that has been picked up, cleaned, and reprocessed. This study provides insight in the macro and microstructural changes produced by several cycles of recycling in these materials. Specimens in the form of film for supermarket bags formed with these polymers have been subjected to several recycling sequences. The process closely mimics industrial processes. Four cycles have been applied to the samples. The evolution of mechanical properties, including tensile strength, elongation at break, and tear and impact tests, shows an obvious decrease due to degradation that is not an impediment for practical use after the four cycles of recycling according to the main specifications defined by the producer. Colorimetric measurements reveal no significant variations in the color of the films. The results of the FTIR and TGA analysis show degradation phenomena and changes in crystallinity in branching and the apparition of crosslinking that are in consonance with the mechanical data. There is also a difference between both types of recycled PE. In general, rLLDPE is more affected by the recycling than rPE. According to our findings, the limiting property would be the tearing. By comparing these values with bags available in the market, manufactured from 70–80% recycled material, we can infer that while two reprocessing cycles can lead to good results, a maximum of four cycles of recycling is advisable.

## 1. Introduction

Plastics are ubiquitous in our daily lives, significantly enhancing our quality of life while fulfilling a myriad of purposes and needs. In 2022, the European region alone witnessed the transformation of 58.7 million tons of plastic [1], with the packaging sector emerging as the foremost consumer, accounting for a 39.6% of total production. This production was primarily based in the production of polypropylene (15.4%) and polyethylene (13.4% LDPE/LLDPE and 8.7% HDPE). The plastic bags industry is a prominent player in the packaging sector.

This significant use of plastics highlights their role in contemporary society, requiring a comprehensive understanding of their life cycle, including critical elements such as recycling and waste management. It also underlines the importance of sustainable practices and innovative solutions in the plastics industry. These measures are indispensable to cope with growing demand and, at the same time, mitigate adverse environmental impacts. Efforts to improve the circular economy of plastics, exemplified by the European Circular Economy Package, have a potential to reconcile multifaceted objectives such as economic growth, resource conservation, and environmental protection [2]. However, these efforts are not enough to significantly reduce the large amount of waste generated each year. The governments of each country and specifically in Europe, the European Union government, are trying to regulate the type, shape, and purpose of material used for each use with explicitly sustainable criteria.

The plastics industry stands as an essential pillar of the European economy, which implies that any strategic action plan must be developed to promote sustainability, efficiency, and competitiveness. The European Circular Economy Package, outlined by the European Commission in 2015, sought to provide this strategic action, especially in sectors with substantial resource use and high circularity potential, such as plastics [2]. The European Green Pact further underlines the urgency, by setting a target of recycling 55% of plastic packaging waste by 2030. In order to achieve this target, packaging will need to include recycled content and designs that encourage reusability, recyclability, and the use of sustainable materials for each specific use (i.e., HDPE netting an not be employed for packaging fruits and vegetables due to its challenging recyclability) [3,4].

Reliable data on production, consumption, waste management, and secondary raw materials would serve as indicators to ensure the success of the transition to circularity, zero emissions, and climate-neutral products. The European Commission attempts to monitor Circular Economy indicators across Europe while reinforcing country-specific national plans [5,6]. The Spanish government, for instance, has issued a decree that compels manufacturers of plastic bags to incorporate a minimum of 70% polyethylene (PE) waste into their products. [7]. Research, innovation, and design are essential to drive change in the plastics industry. It is imperative to recognize the intrinsic value of plastic waste. Mechanical recycling holds great promise due to its simplicity and pre-existing equipment infrastructure. However, the whole process of collection, separation, washing, drying, and extrusion is labor intensive. The ultimate goal is to reintroduce the obtained polymer into the market with properties similar to those of the original product [8].

Because of the considerations cited above, mechanical recycling stands as the most prevalent method, accounting for 42% of plastic packaging waste (PPW) treatment. It offers numerous advantages, including a reduced carbon footprint and the potential to incorporate recycled material into the manufacturing of new packaging products, thereby closing the loop in the plastic circular economy. Nevertheless, it is essential to acknowledge that the thermal and mechanical processes to which the material is subjected result in degradation, leading to molecular structure modifications that impact mechanical, optical, and thermal stability properties. This is particularly pertinent for semi-crystalline polymers like polyethylene (PE) and polypropylene, which are susceptible to thermal and thermo-oxidative degradation, leading to chain breakage and cross-linking [9,10].

Manufactured products, especially those incorporating significant quantities of materials obtained through mechanical recycling processes, must undergo rigorous testing and exhaustive research. This is imperative to ensure the effective transition from virgin material to post-consumer recycled material, and that the resulting product complies with the requisite standards for its intended application in its final function. [11,12]. The necessity for these assessments becomes even more apparent in a context where the demand for recycled materials is consistently on the rise. For instance, the suitability of recycled material for its final function can vary depending on the product type. In the automotive sector, for example, factors such as strength and durability assume critical significance. To guarantee that components produced using recycled material meet the strict quality and safety standards, specific tests assessing structural integrity and long-term performance are essential. In the packaging industry, material specifications can also vary significantly depending on the specific field of application. For instance, the food industry may have requirements that differ greatly from other sectors. In this case, the study has been restricted to the primary activity of the company Sphere, which involves the production of plastic films and bags. This aspect has attracted the interest of numerous researchers [13,14].

Some consumers are growing more inclined to consider the environmental attributes of their chosen products, aligning their purchases with sustainability initiatives. In response, companies are progressively embracing sustainable practices, incorporating eco-friendly materials into their product offerings. This aligns with governmental directives [15], meets consumer demands, and takes into account economic and brand positioning considerations.

As participants in the packaging industry, particularly in the field of plastic bags, our goal is to increase the sustainability of our products. In order to achieve a sustainable, competitive product, we aim to understand the structural changes that take place in low-density linear polyethylene (LLDPE) throughout the actual bag manufacturing process. This study involves the machinery, process conditions, and real-world manufacturing operations. It evaluates the effects of the recycling processes following sequential blown and film extrusion conducted at the Sphere Spain industrial facility. The idea involves simulating the sequence inherent in a recycling cycle. Through the characterization of film produced from material that has undergone multiple recycling processes, we examine variations and structural changes.

This approach is consistent with the manufacturing of films from neat LLDPE and LLDPE recycled post-industry (rLLDPE), which is generated as waste from our own industrial process, but this is not our only objective. We desire to incorporate a substantial amount of post-consumer collected polyethylene (rPEpostc) into our films, which is material that has been retrieved, cleaned, and reprocessed. Utilizing this recycled material adds complexity to the study because rPEpostc is typically more degraded, and the impact of recycling on its structure remains undefined. Our goal is to demonstrate the feasibility of mechanically recycling rLLDPE and rPEpostc, explore blending possibilities with virgin material, and determine the number of recycling cycles possible without compromising the technical specifications of the final product. The macroscopic results are correlated to further investigations by thermal and FTIR characterization on order to provide insight in the distinctive phenomena that take place in each type of material.

## 2. Materials and Methods

### 2.1. Materials

The original, non-recycled linear low-density polyethylene (LLDPE) was supplied by Dow Chemical Iberica (Tarragona, Spain) (Dow 2645.01 G). The material has a density of 0.918 g/cm^3^ and an MFI of 0.85 g/10 min (at 190 °C/2.16 Kg) according to ASTM D792 and ASTM D1238, respectively [16,17,18]. The composition of this grade includes primary antioxidants of the phenol type and secondary compounds, primarily phosphites, designed to prolong the material’s useful life and safeguard it from degradation. The post-consumer and post-commercial PE used to produce the samples of rPEpostc was supplied by the company La Red, Reciclados Plásticos, (Sevilla, Spain) with a density of 0.92–0.94 g/cm^3^ and MFI de 1.15–1.65 g/10 min (190 °C/2.16 Kg). Because of its origin, this material is a mix of several PE types, not only LLDPE. The supplier adjusts the properties by blending different sources.

### 2.2. Circular Economy Applied to Packaging of Plastics

At an industrial scale, Figure 1 illustrates a circular concept of packaging production and the associated recycling processes, with a particular focus on plastic bags. The cycle begins with the manufacturing of packaging elements—films made from raw primary materials, LLDPE in our case, supplied from stage 1. These raw materials are blended with waste of the LLDPE generated during the fabrication process (rLLDPE). The films, in the form of bags, are then utilized by consumers, discarded, collected, and recycled. Post-consumer collected polyethylene (rPEpostc) is incorporated into the materials used to produce the films. This rPEpostc includes various types of polyethylene.

This study specifically focusses into the processes included in stage 2, involving packaging films that have completed the entire cycle. This includes stages involving customers, sorting, selection, and supply from stage 5.

### 2.3. Manufacture of the Samples

The samples were fabricated following the industrial process and machinery applied to obtain real consumer-oriented films. To obtain samples that would allow the assessment of the differences between the use of rLLDPE and rPEpostc, a series of steps are followed:Fabrication of a first reference sample using LLDPE by blown film extrusion;The obtained film is ground and recovered by extrusion forming newly pellets;The pellets are used again in the production of film by blown extrusion;This process is repeated 4 times, analyzing the changes induced in the samples by the repeated recycling of the film.

The purpose of this process is to simulate the impact of including LLDPE waste generated within our own industrial process in the films. This enables a comparison between using raw LLDPE and recycled LLDPE (rLLDPE) that have undergone multiple reprocessing cycles in the properties of the final film.

The same processes are applied to the rPEpostc, utilizing the post-consumer plastic mentioned earlier. It is noteworthy that this product is a blend with an uncertain and variable composition, potentially incorporating different types of PE. The supplier establishes the general properties of this material, maintaining them within established limits by sourcing from various origins. The objective of these second type of samples is to simulate the effects of incorporating the rPEpostc into the films. This facilitates a comparison between using raw LLDPE and rPEpostc, which comprises various types of PE and has undergone multiple reprocessing cycles, in the properties of the final film.

The extrusion process is carried out in a co-extruder Kiefel Kirion N 713220 (Kiefel, Freilassing, Germany). The co-extruder is composed by two extruders of diameters 80 mm and 70 mm and a 26 mm diameter-length model 80/26 F Kirion HEM 2407 mm and 70/26 F Kirion HEM 2082 mm (Kiefel, Freilassing, Germany). Both include various sections of blending converging to a trilayer die head with a diameter of 300 mm and a separation of 2 mm. (Figure 2). The process conditions have remained constant throughout the various with a throughput of 250 Kg/h, polymer temperatures maintained at 210 ± 5 °C, and pressures set at 530 ± 20 bar (A) and 460 ± 30 bar (B) for rLLDPE. For rPEpostc, the pressure values are 260 ± 10 bar (A) and 230 ± 30 bar (B).

The resulting film is structured as a trilayer, with all three layers having identical composition and suitability for shopping bags.

### 2.4. Film Characterization

For every cycle of production/recovery of the material, the mechanical properties of the samples were measured. The tests performed were impact, tensile, and tearing. Given the importance of the visual attributes in films, a colorimetric and gloss study have also been performed. To achieve a better understanding of the microstructural alterations in the material following each cycle, thermogravimetric analysis and Fourier-Transform Infrared spectroscopy (FTIR) have also been carried out on the samples.

The impact properties were evaluated by a dart test according to ISO 7765-1 [19], with Metrotec equipment (Metrotec, Lezo, Spain), the fracture is produced by a dart falling from a specified height. Enough distance was kept between successive tests in order to ensure no influence from previous impacts. The standard technique for impact resistance determination is the “staircase method”. According to this technique, a uniform increase in the dart weight is used during the test, and it is increased or decreased after each impact, depending on the observed result in the test (failure or no failure). The total mass was adjusted following the mentioned method. The conditions were impact radius 8 cm, dart weight 25 g/50 g/100 g, added weights 5 g/15 g/30 g/45 g/60 g/90 g/100 g. A minimum of 10 breaks was required to determine the value.

Tensile strength and elongation at the break of the samples were tested according to the standard ISO 527 [20] considering the orientation of the test tube, using an testing machine manufactured by IDM Test (Ingeniería y Desarrollo de Máquinas, S.L. San Sebastian, Spain) model 0301N208 with cell load capacity of 250 N. Tensile tests were performed at a cross-head speed of 500 mm/min. Direct extension measurements were conducted periodically using an extensometer with sensor arm. A total of 16 replicates of the test are carried out in each direction analyzed, with a constant width of the specimen of 15 mm and a distance of 50 mm between the grips.

The Elmendorf tear properties of all blown films were assessed using an Elmendorf DEA-80 Tear tester manufactured by IDM Test (Ingeniería y Desarrollo de Máquinas, S.L., San Sebastian, Spain) following the standard ISO 6383-2 [21], where two film sections of 76 mm × 63 mm are cut with a sample cutter from each test film produced and their thickness measured. A 20 mm slit was made at the center of the edge perpendicular to the direction being tested. A total of 8 replicates of each test were performed in each direction.

A colorimetric study has also been performed in the samples, originally transparent and colorless. The measurements were made with a PCE-CSM-2 colorimeter (PCE Instruments, Meschede, Germany) determining the tristimulus values of 16 stacked layers of samples on white surface. Tristimulus values measure the light intensity of the primary color values in a sample. Under standardized conditions, this system visually compares colors to red, green, and blue, replicating the perception of the human eye. The outcomes are then represented as coordinates on a graph. To evaluate the differences, the L*a*b* CIELAB color space is taken as reference [22]. The three coordinates of CIELAB represent the lightness (L* = 0 black and L* = 100 white), the balance red/green (a* green = negative; red = positive), and position yellow/blue (b*, blue = negative; yellow = positive). The asterisks (*) after L*, a*, and b* are pronounced star and are part of the full name to distinguish L*a*b* from the Hunter Lab scale. A depiction of the CIELAB color space will be presented in the results discussed below.

The gloss of the samples has been measured by a PCE-PGM 60 (PCE Instruments, Meschede, Germany) that determines the gloss with 8 layers of film obtaining the percentage of the gloss compared to a calibrated reference designated as Gloss 0 or basic gloss.

TGA was performed on a Perkin Elmer TGA 8000 apparatus (Perkin Elmer, Waltham, MA, USA). Samples weighing approximately 10 mg were placed in a corundum dish. The measurement was conducted in the temperature range 30–600 °C and under oxidant atmosphere (30 mL/min), at a heating rate of 20 °C/min. Obtained results are the average of three measurements per sample. Pyris Instrument Managing Software, Version 11 was used to analyze the data.

Chemical structure of compound degraded samples was determined using FTIR analysis performed by means of a Spectrum Two spectrometer from Perkin Elmer (USA). The device had an ATR attachment with a diamond crystal. Spectra were registered at 2 cm^–1^ resolution and 40 scans in the range of 500–3500 cm^−1^, in which the compound signals related to different deformation bands can be observed. SpectrumTM 10 software was used to analyze the data.

## 3. Results

### 3.1. Specifications

These packaging films are subject to market-driven technical demands, which the company has incorporated in the form of the specifications that appear in Table 1. These values are established to guarantee quality assurance and the minimum criteria necessary for each test, thereby confirming the functionality of the end product. The examined films find their final use in the production of checkout bags (supermarket bag). These bags must also follow the directives of the standard UNE EN 53930 [23]. Table 1 resumes the technical requirements specified for the end product.

### 3.2. Mechanical Properties

The mechanical characteristics of the four samples, corresponding to the four cycles applied to the rLLDPE-based samples, as well as the four samples composed of rPEpostc, have been evaluated.

#### 3.2.1. Tensile Strength

The evaluation of tensile strength in both longitudinal and transversal directions is a common technical specification for assessing the quality of final products in films. In recycled films, this property holds particular significance, especially in relation to the final application of the product [24]. The values of the tensile strength are shown in Figure 3. In both directions, there is a decrease in this property with the processing cycles. The longitudinal tensile strength for the rLLDPE decreases a total of 2.5% through the 4 cycles, whereas the rPEpostc decreases 2% after the 4 cycles. It is also noted that the samples including rLLDPE present, as expected, a higher value of tensile strength in the initial reference and the subsequent recycled samples. These results could be anticipated, because in the recycling, the degradation causes the rupture of the macromolecular chains. However, the decrease, especially in the case of the rLLDPE, is not linear, and there is a significant decrease in cycle 2, followed by an increase in the tensile strength in the subsequent cycles. This phenomenon has been observed previously and it is attributed to an increase in branching and a crosslinking produced by the effects of the process, potentially leading to the formation of structures that can contribute to higher strength [25,26,27]. The crosslinking may be induced by the presence of free radicals produced by various sources as indicated by Singh and Sharma [28].

The transversal tensile strength in the transverse direction follows a slightly different evolution. Similar to the longitudinal direction, in rLLDPE and rPEpostc, the reprocessing cycles result in a decrease that is about 8% for rLLDPE, followed by an increase similar to the one found in the longitudinal direction and a decrease of a 13% for rPEpostc. It is noted that the tensile strength of the rPEpostc follows a trend that is similar to the longitudinal.

#### 3.2.2. Elongation at Break

The percentage of elongation at break exhibits significant variations depending on the processing cycles, with distinct trends in both the longitudinal and transverse directions (Figure 4). In the longitudinal direction, for the rLLDPE, the value decreases with the number of cycles. This occurs because the longer chains in the polymer break down, resulting in reduced chain length, molecular weight, and consequently, a decreased elongation at break of the specimen, while the crosslinking effects described previously also impend a longer elongation. Moreover, it can be noticed than the decrease in rLLDPE after cycle 2 is less pronounced than in subsequent cycles, which is in correspondence with the trend observed previously in tensile strength. For the rPEpostc, the tendency is similar, except in the last cycle of processing where the degraded structure leads to a material with an increased elastomeric behavior [29]. Finally, the decreases in elongation at break in the longitudinal sense are an 18% decrease for the rLLDPE and about a 1% decrease for the rPEpostc. The rLLDPE shows always higher values than rPEpostc.

The elongation at break in the transversal sense follows the same trend in the case of the rLLDPE, showing a continuous decrease. For the rPEpostc, there is an increase in elongation as happens with the longitudinal test, but in this case, it is produced in the 3rd cycle. The transversal elongation decreases a final 5% for rLLDPE and about 3% for rPEpostc which presents always lower absolute values. In every sample, the elongation values achieved surpass the technical requirements specified in Table 1.

#### 3.2.3. Elmendorf Tear Strength Test

The obtained data for this property are included in Figure 5. The tear strength is the mechanical property that exhibits the most significant differences as processing cycles progress. For rLLDPE, in the longitudinal direction, the tear strength value decreases with each cycle, eventually falling to a 15% of the initial value (from 182 to 27 N/mm). The same trend is observed for rLLDPE for tear strength in the transverse direction, where the values show a decreasing trend, albeit at a slower pace, decreasing from 290 to 187 N/mm (−35%). The transverse values are higher generally due the incision being perpendicular to the chains, and therefore, a break is more difficult. The transverse tearing implies the breaking of more covalent bonds than in the longitudinal direction [30,31].

Furthermore, the difference between the results in both directions seems to increase with the processing cycles, meaning that the process of recycling affects more in one direction than the other. The initial morphology of the rLLDPE film tends to be a bit oriented but exhibits a quite random distribution, with only a small proportion of crystalline structures in the longitudinal direction. Consequently, in the first initial film, the difference between tear strength in the longitudinal and transverse directions is more similar than in subsequently cycles [32]. The long chains, after the several cycles, are submitted to orientation and can organize themselves according to the drawing direction, imparting an anisotropic behavior. When these chains break through degradation, the tear strength in the longitudinal direction decreases quite rapidly. In the transverse direction, however, the reticulation that has been observed and commented on previously continues linking the broken macromolecular chains resulting in a tear strength that does not decrease so abruptly.

The behavior of the rPEpostc is quite like rLLDPE. The main difference resides in the observation that the rPEpostc retains its resistance better after the processing cycles in both directions. The explanation in this case is completely linked to the one mentioned above. In this case, the rPEpostc is in a certain way equivalent to a rLLDPE that has been degraded previously. It has fewer long chains because its polymeric chains have already been broken and the effect of the crosslinking and branching phenomena that are started during the degradation have a superior effect to the orientation of the chains. This way, the tear strength, being always lower than rLLDPE, is less affected by the recycling.

When comparing the tear strength values to the technical specifications shown in Table 1, it can be observed that samples rLLDPE 3 and 4 fall short of the required 1600 mN in the longitudinal direction. Additionally, sample rLLDPE 4 does not attain the necessary 6000 mN for a typical plastic bag. The usability of the tested materials is restricted by this property, constraining the number of recycling steps they can undergo.

#### 3.2.4. Impact Test

Impact resistance is closely linked to the material’s orientation, as the falling dart imparts biaxial tension on the sample. Therefore, once again, we observe how in the initial cycles, the material exhibits higher isotropy and greater impact resistance [33]. Despite the force being applied biaxially, the impact propagates in the weaker direction. This means that, with the cycle sequence, due to increased longitudinal orientation, the film displays greater weakness in this direction, which is connected to the decrease in longitudinal tear strength, as analyzed in the previous section. Therefore, as material degradation occurs in the sequence of processing cycles, the impact value progressively decreases with each cycle, decreasing from an initial value of 11 to 7 g/µm. These data are represented in Figure 6. As in previous cases, there is a significative difference in the behavior of the rLLDPE in the second processing cycle.

Impact resistance shows minimal variation in the case of rPEpostc. In the first three cycles, a slight increase is observed, followed by a decrease in the last cycle, eventually reaching a value similar to the second cycle. The percentage of variation is approximately 3%. The rationale behind this behavior is the same exposed previously in the case of the tear test. The shorter chains present in the rPEpostc, which are responsible for its low resistance to impact are less affected by the shortening caused by degradation and more influenced by other mechanisms as the crosslinking produced by the successive recycling [34].

### 3.3. Colorimetry and Optical Properties

#### 3.3.1. Colorimetry

The extrusion and recycling processes result in alterations to the material’s structure and the formation of new compounds due to oxidation. These changes may lead to differences in the color. Color is very important in plastic bags, affecting to the perception of the users. To assess these variations, we use the Lab* color space, also known as CIELAB, as defined by the Commission Internationale de l’Éclairage (CIE). This color space is currently one of the most widely adopted systems for evaluating color differences [35].

The changes in the colorimetric parameters for rLLDPE samples are represented in Table 2. The results of the luminosity support the discussions of the results presented before. There is a slight variation in the evolution of the luminosity related to the changes in the microstructure of the rLLDPE in the samples corresponding to the cycles 2, 3, and 4, where most properties experiment a difference in behavior. The value of ‘a’, representing the shift towards red or green, does not exhibit any discernible trend. These values appear random, making it challenging to draw any conclusions. In contrast, the ‘b’ value, indicating the shift towards yellow, displays a pattern towards yellow, which is typical of the degradation processes, implying the formation of colored oxidation groups [10,26].

In the case of rPEpostc (Table 3), a similar trend observed in the case of rLLDPE is followed with some differences. The ‘a’ value (shift from green to red), tends to increase up to cycle 3 and then decreases slightly in cycle 4, indicating a tendency towards the color red that can be attributed to further degradation and increased presence of colored oxidation groups. The ‘b’ value (shift from blue to yellow) shows a typical increase as the cycles progress, similar to what occurs with rLLDPE. As commented before, this is primarily due to material oxidation. Figure 7 shows the evolution of the colorimetric data of the rLLDPE and rPEpostc samples from cycles 1 and 4.

#### 3.3.2. Gloss

The percentage of gloss is another parameter that affects consumer perception and, at the same time, is a consequence of variations in the structure. When the polymer is melted, mixed, and changes its structure during the blowing process, differences in surface characteristics can emerge depending on molecular properties and relaxation time. In the properties studied, there is a difference in the values obtained for cycle 2–3 and the subsequent cycles. Initially, the gloss tends to decrease slightly (rPEpostc) or even increase (rLLDPE), but after cycle 2, there is a progressive loss of gloss in the rLLDPE samples, as can be seen in Figure 8.

After the initial cycles, the chain scission phenomena, the creation of oxidation groups, and reticulation that have been responsible for the changes in the previously studied properties also cause an increase in the roughness of the surface that translates in a scattering of the light and a decreasing of the gloss [36]. These phenomena affect more the rLLDPE than to the rPEpostc, as happens in the other studied properties.

### 3.4. Infrared Spectroscopy Characterization

The characterization of the samples by FTIR provides insight into the transformations that take place at the microstructural level. Figure 9 shows the results within the 1500–1400 cm^−1^ spectral region of the studied samples. The 1474 and 1464 cm^−1^ bands situated in this spectral range correspond to the scissoring vibrations of the -CH_2_- group, and the ratio between these bands is commonly utilized to determine the crystallinity of the samples [37]. To quantify the degree of crystallinity, the formula proposed by Zerbi et al. [38] has been applied, and the results are presented in Table 4.

Figure 10 shows another spectral range, comprised between 1340 and 1400 cm^−1^. In these spectra, the evolution of the 1377 cm^−1^ and the 1369 cm^−1^ bands can be observed. These bands are related, respectively, to the -CH_3_ group (1377 cm^−1^) and to the -CH_2_- group (1369 cm^−1^) [39].

Comparing the evolution of these two bands, there is a decrease in the absorption of the 1369 cm^−1^ band compared to the 1377 cm^−1^ in rLLDPE according to the number of cycles of the samples. This can be attributed to the degradation process that results in a breaking of the polymeric chains. The intensity of the band related to the -CH_3_ group is higher compared to the -CH_2_- group when the number of recycling stages advances, meaning that the chains are shorter and more terminal methyl groups are present [40]. The 1369 cm^−1^ band in rPEpostc samples, composed of HDPE and LDPE waste, is less prominent because it starts from a polymer that has already been degraded from the start, being constituted by shorter chains. The trend is the same even if it is less marked because starts from a lower intensity value of the 1369 cm^−1^ band.

Table 4 illustrates the variation in the rLLDPE and rPEpostc amorphous and crystalline content as a function of the processing steps. Results show relevant discrepancies according to the spectral bands selected for the evaluation of the content in both the amorphous and crystalline phases. The intensity ratio 730/720 cm^−1^ (Figure 11) leads to random results for amorphous phase without the possibility of establishing the structural modifications as a function of process.

As in other studies [41,42], the determination of the crystallinity through these bands 730/720 cm^−1^ seems inadequate for these samples. On the contrary, the bands 1474/1464 cm^−1^ indicate a decrease in crystallinity for the rLLDPE samples as a function of the number of processing steps (from 71% to 59.4%). For the rPEpostc samples, no clear trend is observed because the samples have a significant lower degree of crystallinity, since they come from a heterogeneous blend of PE.

### 3.5. Thermogravimetric Analysis

The thermal behavior (DTGA) of the rPEpostc samples is displayed in Figure 12, where the thermal decomposition temperature (TdT) increases with the number of processing steps (i.e., 460 °C, 465 °C, 490 °C, and 497 °C). In the first 3 processing steps, a very heterogeneous composition with a wide TdT range is observed. As the processing steps progress, the composition of the samples becomes more homogeneous and, in the last processing step, a very homogenized sample composition appears. This homogenization occurs because the sample degrades during the various processing steps. As observed in the FTIR analysis, there is a reduction in molecular size that brings regularity to the existing structures. This way, the interchain interactions are improved and produce the increase in TdT. The previously analyzed results of the mechanical properties (EaB, TS) and the slight increase in crystallinity from 37.6% to 40.7% (FTIR) corroborate this thermal behavior. Crosslinking and branching phenomena are also contributors to these results.

Figure 13 shows the DTGA of the rLLDPE samples. Their thermal behavior is similar to the rPEpostc samples, with a much a more pronounced increase in the TdT. The first processing steps show that, as in the previous case, the samples define a more heterogeneous composition with a very wide TdT range, due to the variability of rLLDPE used. The samples homogenize from the 3rd processing step with a large increase in TdT (i.e., 430 °C, 460 °C, 485 °C, and 505 °C). As a single-component recycling product (LLDPE), it homogenizes faster than rPEpostc samples. This suggests that during the process, as observed by FTIR, there is also chain breakage, homogenization of average molecular weights, conformational restructuring, and increased interchain interactions. Due to the higher initial degree of crystallinity manifested by these samples (71%), the breakage of the macromolecular chains causes a progressive decrease in the degree of crystallinity from 71% to 59.4%. As can be observed, these results are in accordance with a decrease in both longitudinal and transverse tensile properties.

## 4. Conclusions

In this manuscript, we have compared the behavior during the industrial recycling of two types of PE in order to assess the suitability of these materials for their use in commercial shopping bags. The first one is sourced from industrial recycling (rLLDPE) while the second is a material collected after post-consumer use (rPEpostc). When submitting both polymers to several cycles simulating a recycling process, we observe variations in their properties.

The samples including rPEpostc show a lower value for tensile strength in the initial reference and the subsequent recycled samples. The tensile strength in longitudinal and transverse directions decreases through the cycles of recycling. The rPEpostc, starting with a lower value, tends to retain its properties more than rLLDPE, where the decrease is not continuous. After the 2–3 cycles of reprocessing, there is an increase in the tensile strength that is related to crosslinking and branching effects. The tests for elongation at the break also show an expected decrease in value with the number of cycles. In the longitudinal direction, this a direct result of the reduced chain length and molecular weight. The crosslinking described previously does not contribute to a longer elongation, or only has a slight effect in the second cycle. Both polymers exhibit similar behavior, but in the final processing cycles, rPEpostc demonstrates an intensified viscoelastic performance due to its higher overall degradation. The rLLDPE always shows higher values for elongation than rPEpostc. The transversal elongation follows the same trend, in the case of the rLLDPE, showing a continuous decrease and an alteration in the 3rd cycle for the rPEpostc.

The tear strength is the mechanical property that is most affected by the reprocessing. For rLLDPE, in the machine direction, the tear strength value decreases strongly in both the longitudinal and transverse directions. The transverse values are generally higher. This is attributed to the orientation of the chains in the longitudinal direction, where breaking the sample requires the rupture of the covalent bonds within the polymeric chains. The difference in both directions evolves differently due to the variations in the morphology and orientation of the rLLDPE. The behavior of the rPEpostc is quite similar to rLLDPE, which has a lower tear strength, and retains its values more because it has fewer long chains. Consequently, its results are more dependent of the effects of the crosslinking and branching phenomena. Impact resistance also decreases. As in previous cases, there is a small difference in the behavior of the LLDPE in the second processing cycle. Similarly, to the trend described above, rPEpostc is less affected by the reprocessing.

The colorimetry and gloss test show a loss of luminance and gloss in both cases, and a tendency of yellowing is caused by the oxidation and breaking of chains. The effect is less marked for the rPEpostc.

TGA and FTIR techniques enables the monitoring of the microstructural alterations related to the macroscopic properties results. FTIR reveals chain breakages in both sample types, associated with recycling cycles. This effect is more pronounced in rLLDPE samples, as rPEpostc has already undergone degradation. Additionally, the crystallinity of rLLDPE decreases with recycling, while rPEpostc does not exhibit a clear trend, likely due to its initially low degree of crystallinity. The TdT of the samples essentially rises with the number of processing cycles. As these cycles progress, the degradation process leads to the homogenization of structures within the samples, bringing regularity and promoting improved interchain interactions. Crosslinking and branching phenomena are also contributors to the TdT increase.

According to the findings of this study, it can be concluded that the limiting property which undergoes the most significant decrease during processing is tearing. By comparing the obtained values with the reference specifications and the values of bags currently available on the market, manufactured from 70–80% recycled material, we can infer that while two reprocessing cycles can yield satisfactory results, a maximum of three to four cycles of recycling would be advisable. In the case of the rLLDPE, the number of cycles may be even lower. The final recommendation would depend basically on the amount of rLLDPE or rPE post-consumer material included in the formulation of the final product.

## Figures and Tables

**Figure 1 polymers-16-00916-f001:**
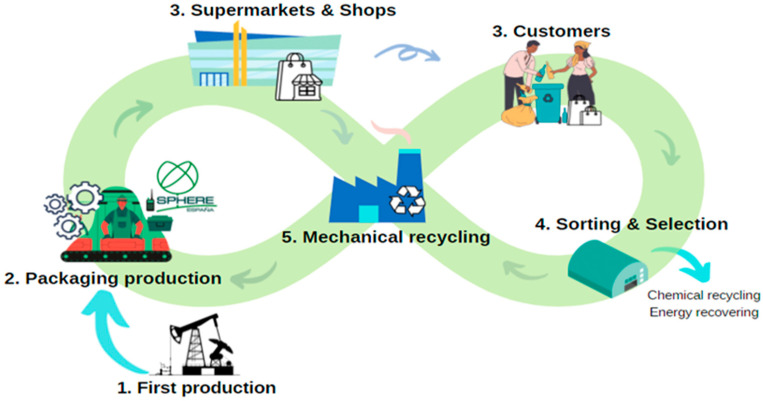
Circular economy applied to the packaging.

**Figure 2 polymers-16-00916-f002:**
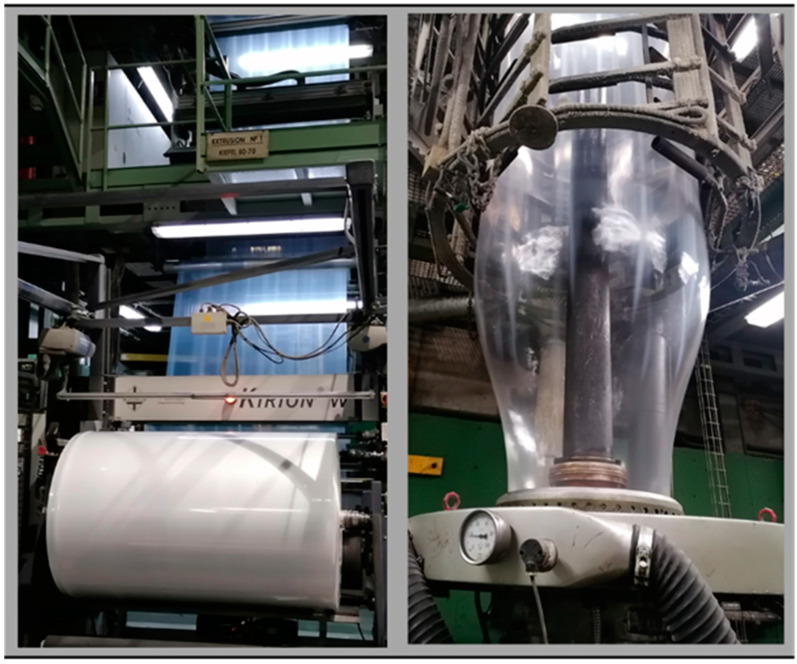
Pictures of the extruder during the production of the film.

**Figure 3 polymers-16-00916-f003:**
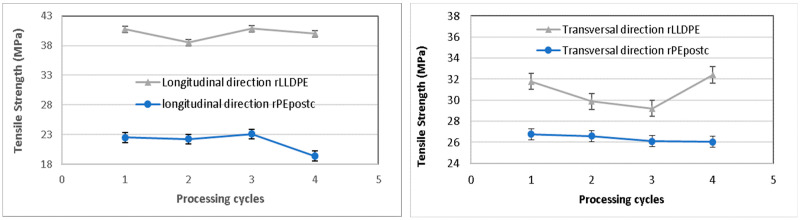
Tensile strength of the samples in longitudinal and transverse directions.

**Figure 4 polymers-16-00916-f004:**
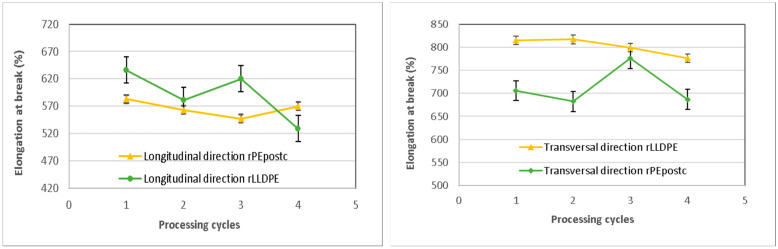
Elongation at break of the samples in longitudinal and transverse directions.

**Figure 5 polymers-16-00916-f005:**
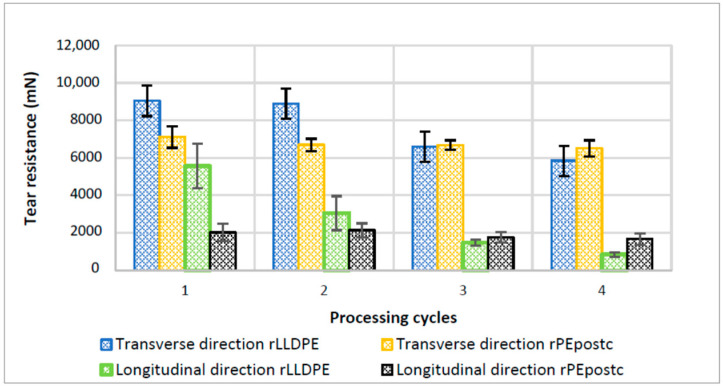
Tear strength test in longitudinal and transversal direction as a function of processing cycles for rLLDPE and rPEpostc.

**Figure 6 polymers-16-00916-f006:**
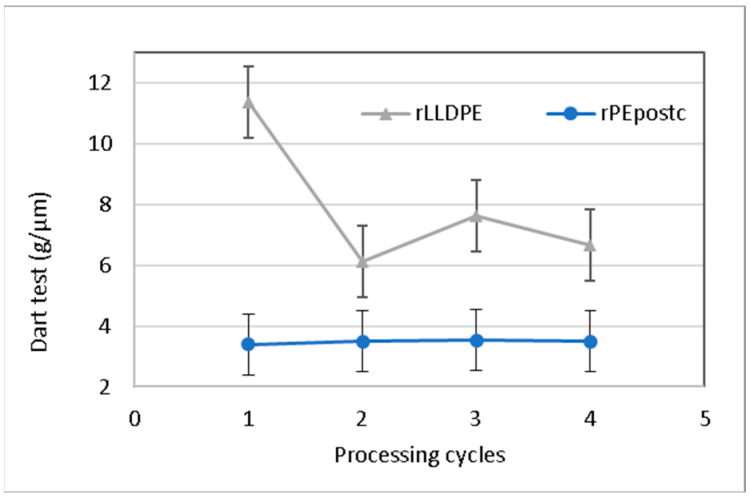
Dart test results as a function of processing cycles for rLLDPE and rPEpostc.

**Figure 7 polymers-16-00916-f007:**
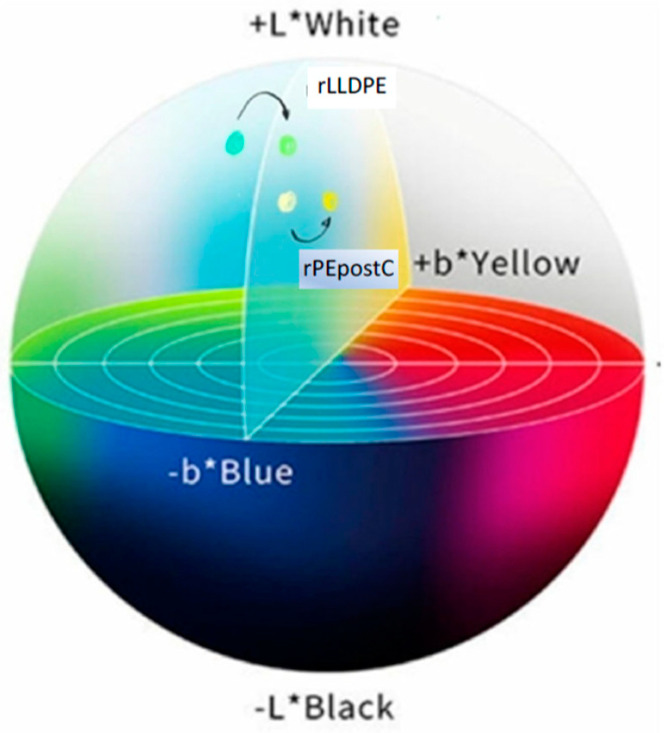
Colorimetric data of the rLLDPE and rPEpostc samples for cycles 1 and 4 (from blue to green for rLLDPE and from white to yellow for rPEpostc).

**Figure 8 polymers-16-00916-f008:**
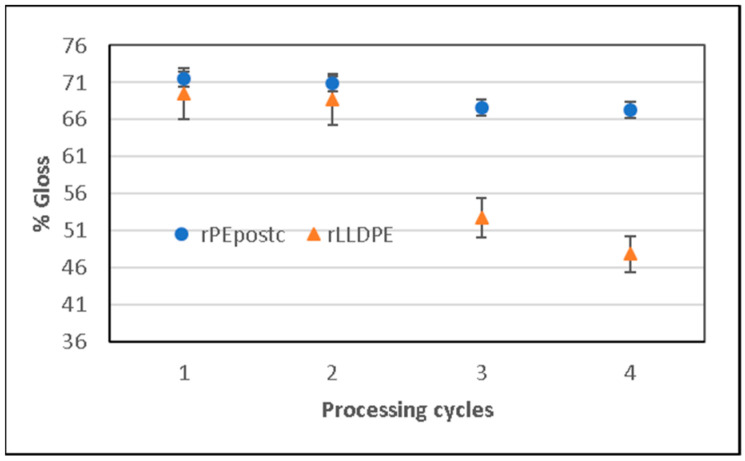
Decreased gloss value for rLLDPE and rPEpostc as a function of processing cycles.

**Figure 9 polymers-16-00916-f009:**
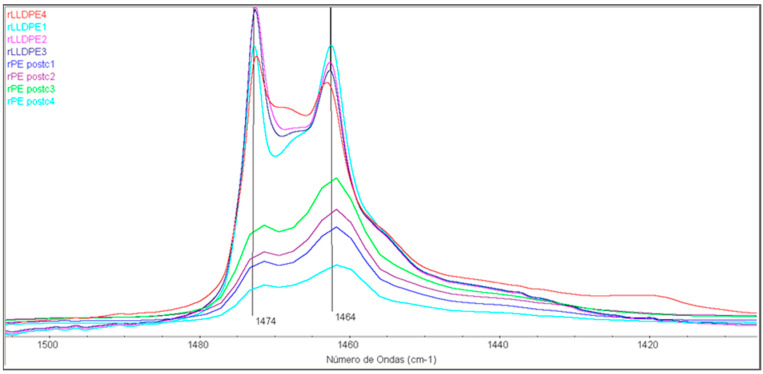
FTIR of the samples in the 1500–1400 cm^−1^ spectral region.

**Figure 10 polymers-16-00916-f010:**
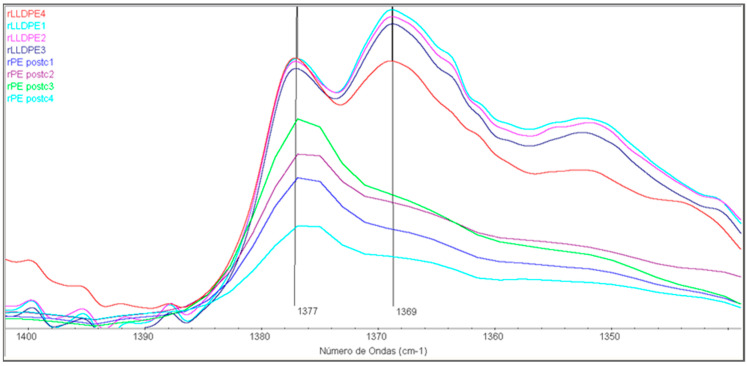
FTIR of the samples in the 1400–1340 cm^−1^ spectral region.

**Figure 11 polymers-16-00916-f011:**
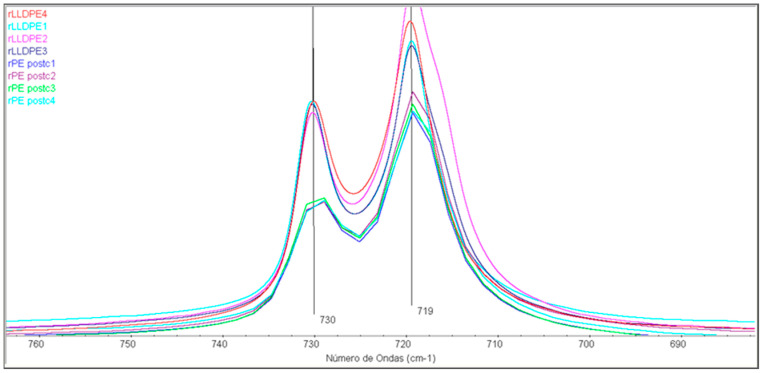
FTIR of the samples in the 760–680 cm^−1^ spectral region.

**Figure 12 polymers-16-00916-f012:**
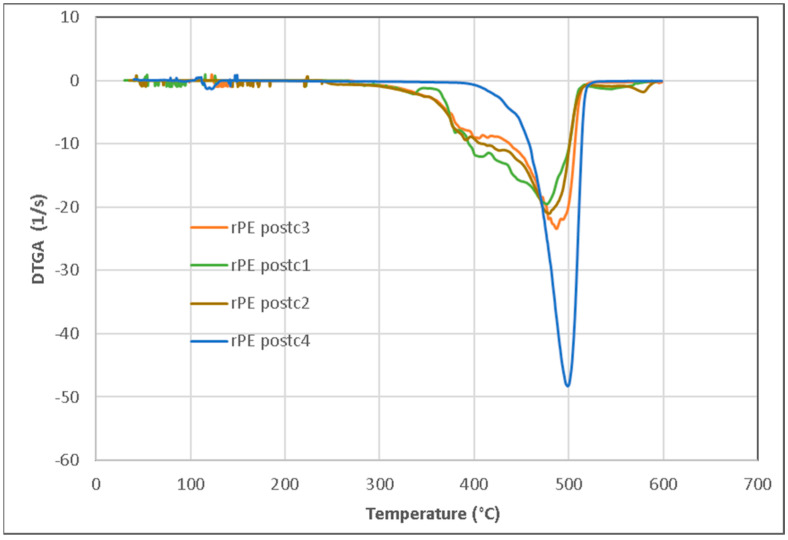
DTGA of the rPEpostc samples.

**Figure 13 polymers-16-00916-f013:**
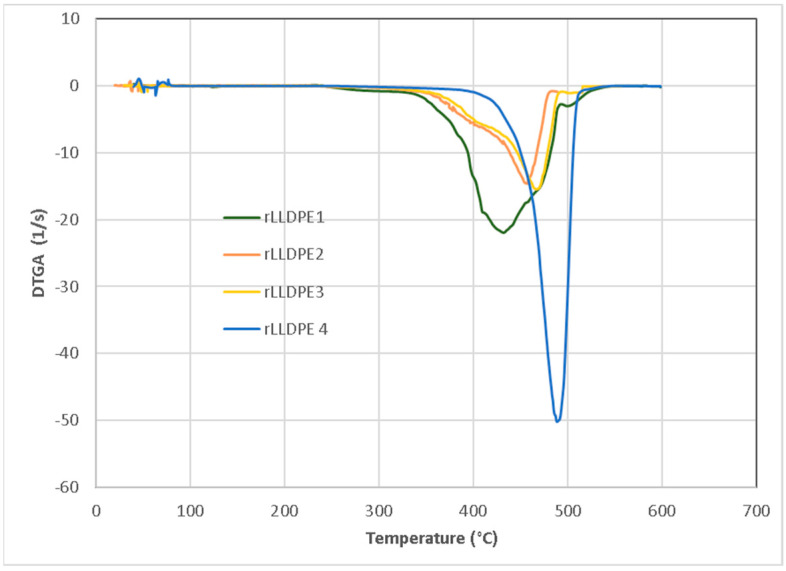
DTGA of the rLLDPE samples.

**Table 1 polymers-16-00916-t001:** Technical requirements for the studied films applied to a checkout bag.

Mechanical Property	Technical Specification—Minimum Value
Impact (g/µm)	3.20
Tensile Strength Longitudinal (MPa)	22
Tensile Strength Transversal (MPa)	18.5
Elongation Longitudinal (%)	400
Elongation Transversal (%)	500
Tearing Longitudinal (mN)	1600
Tearing Transversal (mN)	6000

**Table 2 polymers-16-00916-t002:** Colorimetric data of the rLLDPE samples; L* = luminosity; a* = red–green axis (+a red, −a green); b* = yellow–blue axis (+b yellow, −b blue).

Processing Cycles	L*	a*	b*
1	80.05	−1.74	−2.52
2	81.16	−2.20	−1.49
3	79.51	−1.97	−1.04
4	79.52	−1.82	−0.36

**Table 3 polymers-16-00916-t003:** Colorimetric data of the rPEpostc samples; L* = luminosity; a* = red–green axis (+a red, −a green); b* = yellow–blue axis (+b yellow, −b blue).

Processing Cycles	L*	a*	b*
1	70.14	0.40	5.51
2	69.23	0.57	6.19
3	66.69	1.17	6.49
4	68.36	1.13	6.72

**Table 4 polymers-16-00916-t004:** Amount of amorphous and crystalline phase in the rLLDPE and rPE samples.

Samples	A1474/A1464	%Amorphous	%Crystallinity	A730/A720	%Amorphous	%Crystallinity
rLLDPE1	1.22	28.6	71.4	0.59	71.5	28.5
rLLDPE2	1.26	26.4	73.,6	0.82	52.2	47.8
rLLDPE3	1.10	34.9	65.1	0.73	59.2	40.8
rLLDPE4	1.00	40.6	59.4	0.78	55.3	44.7
rPEpostc1	0.70	62.4	37.6	0.61	69.7	30.3
rPEpostc2	0.69	62.7	37.3	0.54	76.9	23.1
rPEpostc3	0.72	60.5	39.5	0.60	71.4	28.6
rPEpostc4	0.73	59.3	40.7	0.61	69.7	30.3

## Data Availability

Data are contained in the article.

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
