# Peer review of "Comparative Analysis of the Effects of Incorporating Post-Industrial Recycled LLDPE and Post-Consumer PE in Films: Macrostructural and Microstructural Perspectives in the Packaging Industry"

_polymers, 2024, doi:10.3390/polym16070916_

Round 1
Reviewer 1 Report
Comments and Suggestions for Authors
Dear Authors
Your manuscript is quite interesting; the comparative analysis of r-LLDPE and postcPE is well organized and very informative.
Most of the tests and consequently the results are explained in details that could be of potential interest for the people involved in industrial sector of food/polymers as well as in retail shops.
I hope that this study will increase the awareness of people in use of polymeric bags or so.
Discussion of the results is very competent and useful. Conclusions are described in details as well, good work.
Please, see the suggestions provided in the attached file - hopefully it will help in improving your manuscript.
Thank you.

English is fine.
Author Response
Your manuscript is quite interesting; the comparative analysis of r-LLDPE and postcPE is well organized and very informative.
Most of the tests and consequently the results are explained in details that could be of potential interest for the people involved in industrial sector of food/polymers as well as in retail shops.
I hope that this study will increase the awareness of people in use of polymeric bags or so.
Discussion of the results is very competent and useful. Conclusions are described in details as well, good work.
Please, see the suggestions provided in the attached file - hopefully it will help in improving your manuscript.
Thank you.
Thank you for reviewing and providing positive feedback. We have integrated all the suggested changes into the manuscript, marking them in yellow, and we include brief descriptions below. Additionally, we have addressed the errors related to subindexes and other typos as pointed out by the reviewer.
- is there is a ref. to support or someone that observe recycled polymers in terms of their long. and trans. tensile strength?
According to the indications of the reviewer we have included a reference where the tensile strenght in longitudinal and transverse directions are studied in the case of recycled packaging films:
Seier, M.; Archodoulaki, V.-M.; Koch, T.; Duscher, B.; Gahleitner, M. Prospects for Recyclable Multilayer Packaging: A Case Study. Polymers 2023, 15, 2966. https://doi.org/10.3390/polym15132966
- We have rephrased and corrected the sentences that were highlighted by the reviewer due to their lack of clarity.
Reviewer 2 Report
Comments and Suggestions for Authors
The authors presented an article on: Comparative analysis of the effects of incorporating post-in-2 dustrial recycled LLDPE and post-consumer PE in films. Macro-3 structural and microstructural perspectives in the packaging in-4 dustry. The article was prepared in an orderly, logical way and contains all the necessary elements. The authors present a very important issue of sustainable development and support their hypotheses with the presented research results. The introduction requires further development. An additional literature review with the latest knowledge would be very helpful in understanding the presented content. I will not make any further comments.
Comments on the Quality of English LanguageI have no comments
Author Response
The authors presented an article on: Comparative analysis of the effects of incorporating post-industrial recycled LLDPE and post-consumer PE in films. Macro-3 structural and microstructural perspectives in the packaging industry. The article was prepared in an orderly, logical way and contains all the necessary elements. The authors present a very important issue of sustainable development and support their hypotheses with the presented research results. The introduction requires further development. An additional literature review with the latest knowledge would be very helpful in understanding the presented content. I will not make any further comments.
Thanks for your review and positive feedback. In accordance with your suggestions, we have incorporated additional references into the introduction.
Seier, M.; Archodoulaki, V.-M.; Koch, T.; Duscher, B.; Gahleitner, M. Prospects for Recyclable Multilayer Packaging: A Case Study. Polymers 2023, 15, 2966.
O'Rourke K, Wurzer C, Murray J, Doyle A, Doyle K, Griffin C, Christensen B, Brádaigh CMÓ, Ray D. Diverted from Landfill: Reuse of Single-Use Plastic Packaging Waste. Polymers (Basel). 2022 Dec 15;14(24):5485. doi: 10.3390/polym14245485. PMID: 36559852; PMCID: PMC9785204.
Cabrera, G.; Li, J.; Maazouz, A.; Lamnawar, K. A Journey from Processing to Recycling of Multilayer Waste Films: A Review of Main Challenges and Prospects. Polymers 2022, 14, 2319. https://doi.org/10.3390/polym14122319
Meert, J., Izzo, A., Atkinson, John D. Impact of plastic bag bans on retail return polyethylene film recycling contamination rates and speciation, Waste Management,Volume 135,2021,Pages 234-242,ISSN 0956-053X,https://doi.org/10.1016/j.wasman.2021.08.043.
Reviewer 3 Report
Comments and Suggestions for Authors
The aim of the manuscript is to investigate the multiple recycling of PE-LLD from industrial waste and PE-LD from post-consumer waste. The application as a shopping bag is used as a central point.
The effect of recycling was tested on the source materials PE-LLD and PE-LD, the former possibly as post-industrial recyclate, the latter as post-consumer recyclate. The properties of the recyclates were examined primarily in terms of mechanical properties (puncture resistance, stress-strain properties, Elmendorf tensile strength). The color and reflective properties were tested. Furthermore, the thermal properties were examined in order to identify differences in the polymer samples depending on the number of recycling cycles, as were the bonds within the polymer molecules using Fourier infrared spectroscopy.
The aim of the investigations was to examine the usability of mechanically recycled PE-LLD and PE-LD in standard industrial processes for the production of blown films.
In general, the article deserves to be published because, despite the long-known problem and despite many similar publications, the specific question and the specific materials are rarely found in the scientific literature.
However, there are some fundamental points that need to be noted first:
The authors often talk about "packaging" or "commercial products" in general when referring to the application, but without clearly stating that they have focused merely on the application as supermarket shopping bags. However, the selection of methods for material production and the investigations and finally the conclusions clearly refer to this application. This should therefore also be clearly stated in the introduction and objectives. For applications as food packaging, for example, completely different properties would have been important, such as the presence of health-relevant contaminants and odor characteristics, i.e. conformity with food law. However, it must also be said that the odor is also relevant for shopping bags and is therefore missing here as an important property. This should at least be mentioned in the text.
The abstract is also too superficial in this respect. The investigated properties and their relevance for potential applications should be better elaborated here.
The linguistic quality also leaves something to be desired in some points and a revision is certainly necessary. Not many, but some terms have been used misleadingly or even incorrectly. There are also some relics from the original language (e.g. line 145 or line 606).
Now the individual points:
Line 13: A mandatory use of at least 70% post-consumer recyclates in shopping bags must be specified more precisely: From when should this apply and where can you read about it?
Lines 15 / 16: It should be stated more precisely here, both linguistically and factually, what the source materials actually were.
Lines 62 to 66: The current trend in EU legislation is not included here. The following sentence does not correspond to the intentions of the current draft of the PPWR. The entire introduction should be shortened and focus on the object of investigation (shopping bags).
Lines 100 / 101: There is no evidence for this statement.
Lines 149 to 153 and Figure 2: The presentation leaves open the extent to which the different types of packaging on the market and the special application in shopping bags can be combined in the recycling process. If you know the intentions, you will understand the background, but clear wording would be very helpful here.
The entire section 2.3 is too imprecise. It is assumed (but not known for sure, line 166) that PE-LLD was used both as virgin material and as post-industrial recyclate. PE-LD was apparently only used as a post-consumer recyclate from the outset. A uniform approach should actually always be used here, i.e. virgin material as a reference and recyclates with different recycling frequencies as the object of investigation. A three-layer extrusion die was also used, but it was not possible to say whether only a monolayer or a three-layer structure was produced here. This section therefore needs to be fundamentally revised. A table for the reference materials and for the samples with different numbers of repeated extrusions together with their designations would be very helpful.
Lines 198 / 199: Very difficult to understand.
Line 203: Poorly understandable. Where was IDM explained for the first time?
Lines 206 / 207: What type of extensometer was used? Which directions do you mean?
Line 2018: Explain "tristimulus values"!
Lines 219 - 224: Explain the differences between the two different color measurements.
Line 227: What do you mean by "layers of film"?
Chapter 3.2.1: Here, together with the information in Chapter 2, you must provide sufficient clarity as to which samples you have examined in which film direction. The results for the new material (0 processing cycles) should also be included in Figure 4.
Chapters 3.2.2 and 3.2.3: Here too, the chapters should be revised in terms of language and clarity. The same as to Figure 4 applies to Figures 5 and 6.
Line 305: This statement is incorrect. The value falls to 15% of the original value or by 85%.
Line 330: You keep writing about the length of the polymer chains, but have not made a single measurement (for example with GPC).
Line 334 and Table 1: Mention the application for which the technical specifications are intended.
Lines 337 / 338: Aren't shopping bags made from a combination of PE-LD and PE-LLD anyway? Then it is difficult to compare the measured values with the specifications.
Line 354: In view of the lack of significance of the measurement results in Figure 7, I consider this statement to be unsubstantiated.
Chapter 3.3.1: Here I find that it is a methodological weakness that no colored PE LLD bags were tested.
Table 2: Please provide error ranges for the measured values. This would also facilitate the discussion.
Line 383: The table heading is not correct here.
Figure 8: Here the image with its details is not clearly recognizable and the legends are incomplete.
Figure 9: Reference values are also missing here.
Line 409: An additional direct roughness measurement would have been nice, e.g. with an AFM.
Chapter 3.4: The -1 is not superscripted everywhere for the wavenumber, and the subscript is missing for the chemical groups.
Table 4 and rows 441 - 443: Why do you include non-significant results in the results table? It is sufficient to indicate the absence of a significant difference.
Line 457: °C is normally found on every keyboard, see also line 474.
Figure 13: Why this unusual sample sequence? Explain the completely different shape of sample no. 4.
Figure 14: Sample 4 also deviates extremely strongly here. I consider both curves to be questionable.
Line 487: This is not about "commercial products" in general, but about shopping bags. I consider the generalization to be questionable, see above.
Line 502: From this sentence one assumes that you have examined PE-LLD virgin material. However, this is not clear from the entire text.
Lines 506 / 507: What exactly do you mean by this sentence?
see previous section
Author Response
The answer to reviewer 3 is in attached file

Round 2
Reviewer 3 Report
Comments and Suggestions for Authors
I think the manuscript is now quite fine after the substantial corrections.